# TOKEN-LEVEL INFERENCE-TIME ALIGNMENT FOR VISION-LANGUAGE MODELS

## ABSTRACT

Vision-Language Models (VLMs) have become essential backbones of modern multimodal intelligence, yet their outputs remain prone to hallucination-plausible text misaligned with visual inputs. Existing alignment approaches often rely on expensive fine-tuning with annotated preference data or sequence-level inference strategies that provide only coarse, delayed feedback. To overcome these limitations, we present TITA (**T**oken-level **I**nference-**T**ime **A**lignment), a lightweight framework that freezes the base VLM and instead trains a reward model to approximate its distribution. During inference, implicit preference signals are extracted as log-probability ratios between the reward model and the target VLM, yielding dense autoregressive feedback. This formulation can be viewed as an inference-time variant of Direct Preference Optimization (DPO), providing token-level corrective signals without retraining the backbone. Extensive evaluations on LLaVA-1.5-7B and 13B show consistent gains across 12 benchmarks, with improvements of +8.6% on MMVet and +6.7% on POPE, indicating stronger general understanding and reduced hallucinations. Additional experiments on Qwen2.5-VL-7B and DeepSeek-VL2-27.5B show comparable gains, especially in hallucination reduction and VQA accuracy, while incurring negligible inference overhead. Our code is available at: https://anonymous.4open.science/r/TITA-BEC6

## 1 INTRODUCTION

Vision-Language Models (VLMs) have transformed multimodal AI, enabling image captioning, visual question answering (VQA), and instruction following by grounding text generation in visual input (Liu et al., 2024a; 2023; Li et al., 2023c; Wang et al., 2024a; Wu et al., 2024a; Zhang et al., 2024; Zhu et al., 2023; Wu et al., 2024c). Yet despite their broad success, VLMs remain prone to a persistent failure mode: *hallucination*—outputs that are fluent but misaligned with the actual visual input. Such hallucinations not only degrade generation quality but also pose substantial safety and reliability risks for trustworthy multimodal AI deployment (Ye et al., 2023; Zhao et al., 2023; Bai et al., 2024; Huang et al., 2024; Leng et al., 2024; Zang et al., 2025).

At the core of this issue, hallucinations often arise from the dominance of language priors over visual grounding, inherited from large-scale pretraining (Li et al., 2023a; Zhu et al., 2023; Hurst et al., 2024; Shen et al., 2025). When visual signals are weak or ambiguous, models default to text-based statistical patterns, amplifying factual inconsistencies. As a result, addressing hallucinations is therefore a central step toward aligning VLMs with human-centric objectives such as accuracy and trustworthiness. Recent studies have explored alignment strategies to better balance visual grounding and language generation, yet existing solutions still struggle to achieve an effective trade-off between performance, scalability, and practicality. As illustrated in Figure 1, current approaches can be broadly categorized into training-time and inference-time alignment.

Training-time alignment methods leverage supervised fine-tuning or reinforcement learning with human or model-based feedback (Xiong et al., 2024; Zhou et al., 2024b; Kapuriya et al., 2024). While effective, they require large annotation budgets or expensive preference labels from proprietary models, limiting accessibility and scalability. Moreover, retraining is often necessary to adapt to new domains, further increasing costs (Zhao et al., 2024; Favero et al., 2024; Bai et al., 2025).

In contrast, inference-time methods avoid retraining by steering frozen VLMs with external reward models (Cui et al., 2024; Deng et al., 2024; Zhu et al., 2024; Yan et al., 2024; Zhou et al., 2024c).

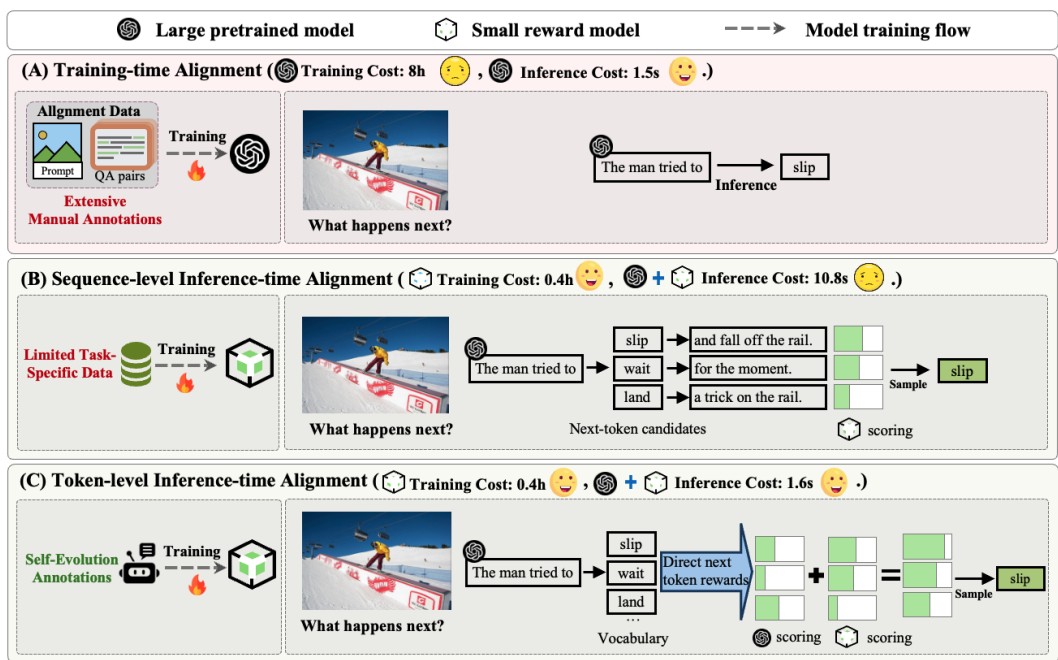

Figure 1: Overview of preference alignment strategies for VLMs (LLaVA-1.5-7B). (A) Training-time alignment fine-tunes base model $\pi_\theta$ with human-labeled preferences. (B) Sequence-level inference-time alignment reranks complete responses with reward models. (C) `TITA` with token-level decoding guidance via implicit preference optimization for lightweight and fine-grained alignment.

Most operate at the sequence level: they assign rewards to entire responses, offering only delayed and coarse-grained feedback while incurring heavy overhead from sampling and reranking. However, this design introduces two critical drawbacks. First, reward signals are delayed and coarse-grained, providing no guidance during intermediate decoding steps where hallucinations typically emerge. Second, evaluating full sequences for each candidate substantially inflates inference costs. Thus, despite progress, hallucination reduction remains expensive and insufficiently fine-grained.

**Intuition and Motivation.** We argue that hallucinations originate not only from weak visual grounding but also from the lack of timely alignment signals during generation (Sun et al., 2023; Li et al., 2024). Sequence-level feedback arrives only after hallucinations have already manifested. By contrast, token-level guidance can intervene earlier, providing fine-grained signals at each decoding step to suppress hallucinations before they propagate. Inspired by prior work (Fu et al., 2024), we further observe that preference information need not rely on costly human annotation or explicit reward models: it can be implicitly captured through log-probability ratios between reference and target models, enabling lightweight preference estimation without retraining.

**Our Approach.** Motivated by these observations, we introduce `TITA` (Token-Level Inference-Time Alignment), a lightweight framework that mitigates hallucinations by transforming sparse sequence-level feedback into dense, autoregressive signals. Instead of fine-tuning the base VLM, it compares token-level probability distributions between a reward model and the target VLM, deriving implicit preferences via log-probability ratios without human annotations or handcrafted rewards. A token-mapping mechanism ensures compatibility across heterogeneous tokenizers, enabling plug-and-play inference-time alignment for off-the-shelf VLMs without modifying their parameters (Figure 1(C)).

In this paper, we establish `TITA` as a general token-level preference-alignment strategy that suppresses hallucinations in VLMs without explicit VLM finetuning, or manually annotated token-level data. Theoretically, we prove that `TITA` can approximate any dense reward distribution over token sequences, bridging the gap between coarse sequence-level and fine-grained token-level alignment (Section A). Methodologically, we design a self-supervised preference construction pipeline that leverages augmented visual inputs to generate robust token-level reward signals without human labels (Section 3.1). Empirically, we conduct extensive evaluations across three representative VLM

families and 12 benchmarks, where `TITA` consistently reduces hallucinations while preserving base model capabilities and incurring minimal computational overhead (Section 4.2).

## 2 RELATED WORK

**Hallucination in VLMs.** VLMs have demonstrated impressive performance across a wide range of multimodal tasks by leveraging the extensive world knowledge of LLMs and the visual perception capabilities of pretrained image encoders (Li et al., 2023c; Liu et al., 2024a; 2023; Wang et al., 2024a; Chen et al., 2024b; Zang et al., 2025). Due to the imbalance in model capacity and data scale between modalities during pretraining, VLMs often exhibit a bias toward language priors, which can lead to hallucinations—fluent yet visually inconsistent or factually incorrect outputs (Bai et al., 2024; Huang et al., 2024; Leng et al., 2024). This compromises factual accuracy and limits deployment in high-stakes applications like healthcare and scientific reasoning (Chen et al., 2024a; Sun et al., 2024; Wu et al., 2024b). Mitigating hallucination has therefore become a central research challenge. Prior efforts (Li et al., 2023a; Yu et al., 2024; Sun et al., 2025) have focused on aligning VLM outputs with human preferences to improve factual consistency and enhance trustworthiness.

**Preference Alignment in VLMs.** Recent efforts aim to align VLMs with human preferences via training-time or inference-time strategies. Training-time alignment involves supervised fine-tuning or reinforcement learning based on human-annotated (Sun et al., 2023; Guo et al., 2025; Shen et al., 2025) or model-generated preference data (Ren et al., 2024; Zhang et al., 2025; Wan et al., 2025). These approaches often yield strong performance but require substantial computational resources and repeated retraining when adapting to new tasks or preferences. In contrast, inference-time alignment introduces external reward models to guide generation from frozen VLMs, avoiding full model updates. While more flexible, most existing inference-time methods operate at the sequence level (Gou et al., 2024; Dong et al., 2025; Sun et al., 2025), computing rewards over entire responses. This coarse-grained feedback delays correction of intermediate errors and increases inference latency. Moreover, simulating full candidate completions per decoding step adds significant overhead.

**Data Augmentation in VLMs** Although data augmentation is ubiquitous in vision tasks (Grill et al., 2020; He et al., 2020), its effects (Chen et al., 2024c; Yuan et al., 2024) on VLMs are considerably less stable: even subtle perturbations can induce semantic shifts and degrade output consistency. Rather than treating this as noise, recent work leverages this property to mine preference pairs from divergent outputs (Awais et al., 2025; Yu et al., 2023b). This turns augmentation into a tool for weak supervision, enabling preference-based training without costly human labels.

**Self-Evolution Strategies.** To further reduce reliance on costly human annotations, self-evolution has emerged as an effective paradigm where models generate their own alignment signals. Approaches such as self-consistency ranking, feedback distillation, and preference mining have been explored in LLMs (Chen et al., 2024c; Patel et al., 2024; Wang et al., 2024b; Ding & Zhang, 2025). Self-evolution has been mostly explored in language-only settings, while its application to VLMs remains limited. `TITA` extends this paradigm by introducing token-level, self-generated preference signals under visual grounding constraints, enabling effective modality alignment with efficiency and scalability.

## 3 METHODS

In response to the inherent tendency of aligned VLMs to develop shallow heuristics rather than principled reasoning, we present a token-level preference optimization framework that fundamentally rethinks the alignment process.

### 3.1 PREFERENCE DATASET CONSTRUCTION

In preference optimization, the dataset is a collection of quadruplets $\mathcal{D} = \{(q_n, I_n, y_w^n, y_l^n)\}_{n=1}^N$, where $q_n$ is the input question, $I_n$ is the associated image, $y_w$ is the preferred response, and $y_l$ is the less preferred one. Preferences are modeled with the Bradley–Terry (BT) formulation:

$$p(y_w \succ y_l | q, I) = \frac{\exp(r(q, I, y_w))}{\exp(r(q, I, y_w)) + \exp(r(q, I, y_l))}, \quad (1)$$

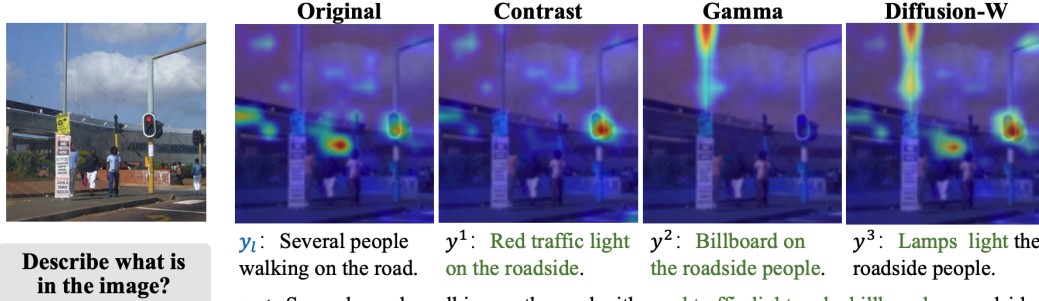

Figure 2: Attention visualization demonstrating how TITA enables holistic caption generation. The winner answer $y_w$ is generated by fusing multiple responses obtained from augmented versions of the image, capturing more comprehensive and details compared to the original generation $y_l$.

where $r(q, I, y)$ is the reward score for response $y$ conditioned on the input $(q, I)$. This formulation naturally captures our intuition that the winning answer should have a higher probability of being preferred, while maintaining a meaningful comparison with the competitive loser.

To construct more informative preference pairs, we leverage the diversity of model outputs generated under multiple image augmentations. Given an input $(q, I)$, we first obtain a baseline response from the original image:

$$y_l \leftarrow \pi_\theta(\cdot|q, I), \tag{2}$$

$$\hat{y}^k \leftarrow \pi_\theta(\cdot|q, f_k(I)), \quad k \in [1, ..., K], \tag{3}$$

$$y_w \leftarrow \pi_\theta(\cdot|\hat{y}^1\|\hat{y}^2\| \ldots \|\hat{y}^K), \tag{4}$$

where $f_k$ denotes the $k$-th image augmentation method, and $y_l$ serves as the *loser* response. The responses $\{\hat{y}^1, \hat{y}^2 \ldots, \hat{y}^K\}$ are concatenated along with a fusion prompt (e.g., "Please provide a comprehensive fusion based on the following candidate answers."), and passed back into the model to generate a unified answer $y_w$, which serves as the *winner* response. This encourages alignment with responses that aggregate diverse visual cues across augmentations.

Figure 2 illustrates how different augmentations highlight distinct visual cues and lead to semantically richer descriptions. The fused output captures fine-grained elements (e.g., red traffic light, billboard) that are overlooked in the original response, validating the effectiveness of our augmentation-guided preference construction.

## 3.2 TOKEN-LEVEL REWARD MODEL

Let $y = (y_1, y_2, \ldots, y_t)$ denote the output token sequence, where $y_t$ is the token at position $t$, and $y_{<t}$ is the prefix. Then the autoregressive reward model assigns token-level rewards by modeling the log-likelihood of each token conditioned on the input and its prefix:

$$r(q, I, y) = \sum_t \pi_r(y_t|q, I, y_{<t}), \tag{5}$$

where $\pi_r(y_t|q, I, y_{<t})$ is a learnable distribution function. Generating the next token requires only one forward pass through the target and reward models. This is significantly faster than previous methods that require generating several candidate tokens, completing the full response for each, and then selecting the best next token. And we prove that this parameterization is sufficiently expressive to guide target LLMs to any distribution achievable by traditional reward models within the KL-regularized RL framework in Appendix A.

Unlike sequence-level reward models (Zhang et al., 2025), which compute next-token rewards by generating full responses following each next-token candidate and then evaluating them with the sequence-level reward model, our approach avoids this computational burden.

Training reward model on a preference dataset involves predicting token-level reward to ensure the sequence-level rewards align with the data, using a negative log-likelihood loss function as follows:

$$\mathcal{L}(\pi_r; \mathcal{D}_p) = -\mathbb{E}_{\mathcal{D}_p}\left[\log \sigma\left(\beta \sum_t \log \pi_r(y_{w,t}|q, I, y_{w,<t}) - \beta \sum_t \log \pi_r(y_{l,t}|q, I, y_{l,<t})\right)\right], \tag{6}$$

## 3.3 INFERENCE-TIME GUIDANCE

In this section, we present our auto-regressive inference-time alignment method. In practical scenarios, fine-tuning a smaller, typically weaker language model (e.g., 1B/7B) is often feasible, while fine-tuning a larger, stronger model (e.g., 70B) may be impractical due to resource constraints. By leveraging our proposed auto-regressive reward model, which predicts next-token rewards $\log \pi_r(y_t|q, I, y_{<t})$ in a manner similar to how language models predict next-token log probabilities, Equation 7 can be interpreted as a form of controlled decoding from multiple models:

$$\log \pi(y|q, I) = -\log Z(q, I) + \sum_t \log \pi_\theta(y_t|q, I, y_{<t}) + \lambda \cdot \sum_t \log \pi_r(y_t|q, I, y_{<[t]}), \quad (7)$$

This formulation allows TITA to apply previous decoding techniques (Dekoninck et al., 2023) to sample the next token $y_t$, conditioned on the query with image $(q, I)$ and the partially generated response $y_{<t}$, by computing the next-token conditional probability as follows:

$$\pi(y_t|q, I, y_{<t}) \propto \pi_\theta(y_t|q, I, y_{<t})\big(\pi_r(y_t|q, I, y_{<t})\big)^\lambda. \quad (8)$$

---

**Algorithm 1** Token-level Inference-time Alignment

**Require:** Dataset with query prompts and images: $\mathcal{D} = \{(q_n, I_n)\}_{n=1}^N$; target model $\pi_\theta$; target model tokenizer $\mathcal{T}_\theta$; reward model $\pi_r$; reward model tokenizer $\mathcal{T}_r$; alignment hyper-parameter $\beta$; inference query prompt and image: $(q^*, I^*)$; number of output tokens $T$; scaling factor $\lambda$; Image augmentation methods $\{f_k(\cdot)\}_{k=1}^K$, $\mathbb{P}$ is the softmax-derived token probability distribution.
1: $\mathcal{D}_p \leftarrow \{\}$ // Construct preference dataset $\mathcal{D}_p$ for reward model training.
2: **for** $n = 1, \ldots, N$ **do**
3:     **for** each augmentation methods $f_k(\cdot)$ **do**
4:         $I_n^k \leftarrow f_k(I_n)$    // Augment images.
5:         $\hat{y}_n^k \sim \pi_\theta(\cdot|q_n, I_n^k)$   // Generate candidate response from augmented input.
6:     **end for**
7:     $y_l^n \sim \pi_\theta(\cdot|q_n, I_n)$    // Loser response generated by the pretrained model.
8:     $y_w^n \sim \text{Fusion}(\hat{y}_n^1, \hat{y}_n^2, \ldots, \hat{y}_n^K)$ // Winner response generated from fusion candidate answers.
9:     $\mathcal{D}_p \leftarrow \mathcal{D}_p \cup (q_n, I_n, y_w^n, y_l^n)$   // Adding the triplet to the preference dataset.
10: **end for**
11: // Training the auto-regressive reward model $\pi_r$.
12:

$$\min_{\pi_r} -\mathbb{E}_{(q,I,y_w,y_l)\sim\mathcal{D}_p}\Big[\log\sigma\Big(\beta\sum_t\log\pi_r(y_{w,t}|q, I, y_{w,<t}) - \beta\sum_t\log\pi_r(y_{l,t}|q, I, y_{l,<t})\Big)\Big]$$

13: // Token-level reward guidance during inference stage.
14: **for** $t = 0, \ldots, T-1$ **do**
15:     **if** $\mathcal{T}_r \neq \mathcal{T}_{\text{target}}$ **then**
16:         $\mathbb{P}[\mathcal{T}_r(\mathcal{V})] \leftarrow \pi_r(y_t|q^*, I^*, y_{<t})$
17:         // Logits mapping with top-$k$ tokens.
18:         $\mathcal{V}^{(k)} \leftarrow$ top-$k$ tokens with highest likelihood
19:         $\mathbb{P}[\mathcal{T}_\theta(\mathcal{V}^{(k)})] \leftarrow \mathbb{P}[\mathcal{T}_r(\mathcal{V}^{(k)})]$
20:         $\pi_{\text{decode}}(y_t|q^*, I^*, y_{<t}) \leftarrow \pi_\theta(y_t|q^*, I^*, y_{<t})\big(\mathbb{P}[\mathcal{T}_\theta(\mathcal{V}^{(k)})]\big)^\lambda$
21:     **else**
22:         $\pi_{\text{decode}}(y_t|q^*, I^*, y_{<t}) \leftarrow \pi_\theta(y_t|q^*, I^*, y_{<t})\big(\mathbb{P}[\mathcal{T}_r(\mathcal{V})]\big)^\lambda$
23:     **end if**
24:     // Next predict token sampling:
25:     $y_t \leftarrow$ top-1 token from logits $\pi_{\text{decode}}(y_t|q^*, I^*, y_{<t})$
26:     $y_{<t+1} \leftarrow y_{<t} \,\|\, y_t$
27: **end for**
**Ensure:** Generated response $y_{<t}$

---

Unlike training the reward model with DPO, where the reference policy (i.e., the target LLM) must be pre-specified during training, TITA trains the autoregressive reward model without relying on

any specific target LLM during the training phase. This design allows the trained autoregressive reward model to be flexibly paired with different target LLMs during the inference stage, providing significant configurability. For instance, a smaller autoregressive reward model can guide a larger target LLM for weak-to-strong alignment. The key distinction lies in inference-time flexibility: DPO ties alignment to a specific target LLM chosen during training, whereas `TITA` decouples reward model training from the target LLM, enabling diverse and adaptable inference-time applications.

We illustrate the complete pipeline of `TITA` in Algorithm 1. After alignment with Equation 6, in each token generation step, if the reward model $\pi_r$ and the target model $\pi_\theta$ have different tokenizers, we need to map the logits of $\pi_r$ to the logits of $\pi_\theta$. When mapping logits, we decode the top-$k$ tokens with the highest probability from $\pi_r(y_t|q, I, y_{<t})$, and then use the tokenizer of the target model to encode these tokens and assign the corresponding probabilities. According to Equation 8, we obtain the output of the target model guided by the reward model. We select the token with the highest probability and repeat this process to generate the complete output.

## 4 EXPERIMENTS

### 4.1 SETTINGS

**Implements Details.** To align with previous preference-based approaches on hallucination mitigation, we take LLaVA-1.5-7B and 13B as the backbone models to validate the effectiveness of `TITA`. To evaluate the effectivenss of `TITA` on more advanced and powerful model, we implement `TITA` based on Qwen2.5-VL-7B-Instruct (Bai et al., 2025) and DeepSeek-VL2-27B (Wu et al., 2024c). And we use TinyLLaVA-1.5B (Zhou et al., 2024a) as the small reward model (Note that the source data obtained from the LLaVA665k SFT dataset (Liu et al., 2024a)). Specifically, image-question pairs from OCRVQA (Mishra et al., 2019) and TextVQA (Singh et al., 2019) (collectively referred to as "text+ocr") within LLaVA665k are used to generate the DPO preference data. Following the settings of prior work (Liu et al., 2024a; Zhao et al., 2023), we take CLIP-VIT-L-336px as the vision encoder, the batch size is 128, and the learning rate is $2e^{-6}$. The default LoRA rank is set to 1024 and the scale parameter $\beta$ in DPO is fixed at 0.1.

**Baselines.** We compare `TITA` with both training- and inference-time preference alignment methods. The training-time methods include Fact-RLHF, CSR, and SeVa. Fact-RLHF (Sun et al., 2023) employs reinforcement learning from human feedback to optimize the base model. CSR (Zhou et al., 2024c) proposes a calibrated self-rewarding strategy that iteratively improves the model by leveraging internally generated reward signals. SeVa (Zhu et al., 2024) also uses DPO for alignment but is limited by its reliance on comparisons between raw and enhanced visual outputs, restricting its ability to model deep semantic preferences. As for inference-time alignment, we consider Critic-V (Zhang et al., 2025), which adopts a Reasoner-Critic architecture: the Reasoner generates reasoning paths based on visual content and corresponding queries, while Critic offers real-time feedback to refine these reasoning trajectories. See the Appendix B.2 for more detailed methods.

**Evaluation Benchmarks.** We evaluate `TITA` using three categories of benchmarks: (1) *Comprehensive Evaluation*: SEED (Li et al., 2023b), LLaVA-Bench (Liu et al., 2024b), MMbench (Liu et al., 2025), MME (Yin et al., 2023), MMVet (Yu et al., 2023a). (2) *General Visual Question Answering (VQA)*: VisWiz (Gurari et al., 2018), GQA (Hudson & Manning, 2019), ScienceQA (Lu et al., 2022), MMStar (Chen et al., 2024b). (3) *Hallucination Detection*: CHAIR (Rohrbach et al., 2018) and POPE (Li et al., 2023d). More detailed information in Appendix B.1.

Table 1: Training cost and configurations of alignment methods evaluated on LLaVA-1.5-7B. For inference-time methods, cost refers to the training time of the reward model.

| Methods | Alignment Stage | Optimization | Dataset | Training Target | Cost |
|---|---|---|---|---|---|
| Fact-RLHF (Sun et al., 2023) | Training-time | RLHF | Human-annotated | Pretrained Model | 16.4h |
| CSR (Zhou et al., 2024c) | Training-time | DPO | Self-generated | Pretrained Model | 6.8h |
| SeVa (Zhu et al., 2024) | Training-time | DPO | Self-generated | Pretrained Model | 7.5h |
| Critic-V (Zhang et al., 2025) | Inference-time (Seq-L) | DPO | GPT-annotated | Reward Model | 2.9h |
| `TITA` (Ours) | Inference-time | DPO | Self-generated | Reward Model | 0.4h |

Seq-L: Sequence-level reward, used to rank the score of each answer with a finetuned critic (reward) model.

Table 2: Comparison of `TITA` and competing alignment methods on LLaVA-1.5-7B and 13B models across vision-language evaluation benchmarks. ↓ indicates lower is better.

| Model | MME$^P$ | MME$^C$ | SEED | LLaVA$^W$ | MMVet | MMB | SQA | GQA | VisWiz | CHAIR$_s$ ↓ | CHAIR$_i$ ↓ | POPE |
|---|---|---|---|---|---|---|---|---|---|---|---|---|
| *Base Model: LLaVA-1.5-7B* | | | | | | | | | | | | |
| Base | 1510.7 | 348.2 | 58.6 | 63.4 | 30.5 | 64.3 | 66.8 | 62.0 | 50.0 | 48.8 | 14.9 | 85.9 |
| + Fact-RLHF (Sun et al., 2023) | 1490.6 | 335.0 | 58.1 | 63.7 | 31.4 | 63.4 | 65.8 | 61.3 | 51.7 | 38.7 | 11.3 | 81.5 |
| + CSR (Zhou et al., 2024c) | 1524.2 | 367.9 | 60.3 | 71.1 | 33.9 | 65.5 | 70.7 | 62.3 | 54.1 | 21.0 | 6.0 | 86.8 |
| + SeVa (Zhu et al., 2024) | 1531.0 | 369.2 | 65.8 | 72.2 | 37.2 | 65.7 | 67.5 | 60.7 | 51.5 | 20.5 | 5.8 | 86.7 |
| + Critic-V (Zhang et al., 2025) | 1528.4 | 355.0 | 63.4 | 67.8 | 35.7 | 64.0 | 66.5 | 59.4 | 51.0 | 26.8 | 7.9 | 86.5 |
| + `TITA` (Ours) | **1538.4** | **369.5** | **66.6** | **72.5** | **39.1** | 65.5 | **70.7** | **62.3** | **54.8** | 20.3 | **5.6** | **91.7** |
| *Base Model: LLaVA-1.5-13B* | | | | | | | | | | | | |
| Base | 1531.3 | 295.4 | 61.6 | 70.7 | 35.4 | 67.7 | 71.6 | 63.3 | 53.6 | 48.3 | 14.1 | 85.9 |
| + Fact-RLHF (Sun et al., 2023) | 1494.2 | **310.4** | 60.7 | 64.9 | 32.6 | 64.7 | 68.2 | 62.8 | 54.5 | 41.2 | 13.7 | 86.7 |
| + CSR (Zhou et al., 2024c) | 1530.6 | 303.9 | 62.9 | 74.7 | 37.8 | **68.8** | **75.1** | 63.7 | **56.8** | 28.0 | 7.3 | 87.3 |
| + SeVa (Zhu et al., 2024) | 1533.9 | 305.1 | **68.6** | 80.1 | 41.0 | 68.7 | 71.2 | 63.4 | 54.7 | 23.6 | **6.5** | 87.4 |
| + Critic-V (Zhang et al., 2025) | 1529.5 | 307.1 | 64.1 | 68.8 | 39.2 | 66.7 | 67.0 | 60.2 | 52.5 | 26.0 | 7.4 | 80.1 |
| + `TITA` (Ours) | **1540.0** | 309.5 | **68.6** | **80.5** | **42.3** | 68.2 | 71.8 | **63.9** | 55.2 | 23.5 | 6.6 | **92.6** |

## 4.2 COMPARISON WITH STATE OF THE ART

**Better efficiency.** Table 1 shows the extremely low training cost of `TITA`. Compared with training-time alignment, such as Fact-RLHF (Sun et al., 2023), CSR (Zhou et al., 2024c), and SeVa (Zhu et al., 2024), `TITA` only needs to train the small reward model (only 1.5B in our experiment setting). Compared with sequence-level inference-time alignment, such as Critic-V (Zhang et al., 2025), `TITA` does not need to rank each answer, but directly assists the pretrained model to infer the next token, which greatly improves efficiency.

**Better effectiveness.** To comprehensively evaluate the effectiveness of our proposed alignment strategy, we compare `TITA` with several SOTA baselines. The results in Table 2 illustrate that `TITA` consistently outperforms baseline models across multiple benchmarks, highlighting its strengths in various vision-language tasks. Across MMVet and MMBench, `TITA` achieved superior overall scores regardless of model size, with specific scoring details in the Appendix 5. In the 7B setting, it attains an "All" score of 39.1% on MMVet, surpassing SeVa (37.2%) and CSR (33.9%). This trend continues in the 13B setting, where `TITA` maintains its lead with an "All" score of 42.3%. The consistently better performance across different scales suggests that the proposed alignment strategy is not only effective but also scalable, offering robust enhancements to the model's comprehension abilities as capacity increases. Further analysis in Appendix B.3 further shows that `TITA` explicitly strengthens visual grounding in the middle layers, thereby mitigating hallucination by preventing the model from over-relying on linguistic priors.

**Generality to recent VLMs.** To examine whether the effectiveness of `TITA` extends beyond LLaVA, we further evaluate it on more recent LVLMs, including Qwen2.5-VL-7B-Instruct and DeepSeek-VL2-27B. For comparison, we adopt Critic-V (Zhang et al., 2025) as the representative sequence-level inference-time alignment baseline, since it is among the most competitive and widely adopted decoding strategies in recent literature. As shown in Table 3, while Critic-V substantially improves

Table 3: Performance of `TITA` on recent VLMs.

| Model | Inference Time | CHAIR$_s$ ↓ | CHAIR$_i$ ↓ | POPE | MMVet |
|---|---|---|---|---|---|
| *Base Model: Qwen2.5-VL-7B-Instruct* | | | | | |
| Base | 1.2s | 37.1 | 9.4 | 91.3 | 61.8 |
| + Critic-V | 7.9s | 18.1 | 6.0 | 95.9 | 64.4 |
| + `TITA` (Ours) | 1.4s | 10.5 | 3.8 | 96.1 | 65.0 |
| *Base Model: DeepSeek-VL2-27B* | | | | | |
| Base | 3.9s | 41.3 | 11.7 | 88.8 | 52.8 |
| + Critic-V | 23.5s | 16.7 | 8.3 | 94.1 | 56.0 |
| + `TITA` (Ours) | 4.2s | 12.5 | 4.9 | 94.7 | 57.3 |

alignment at the cost of high inference latency, `TITA` achieves even stronger hallucination reduction and VQA gains with negligible overhead. These results demonstrate that token-level reward guidance not only generalizes well to modern VLMs but also provides a more efficient alternative to state-of-the-art sequence-level inference-time methods.

**Comparison with alternative decoding methods.** We also compare `TITA` with representative inference-time decoding methods, including VCD (Leng et al., 2024), M3ID (Favero et al., 2024), and MARINE (Zhao et al., 2024). While these approaches adjust logits through heuristic probability

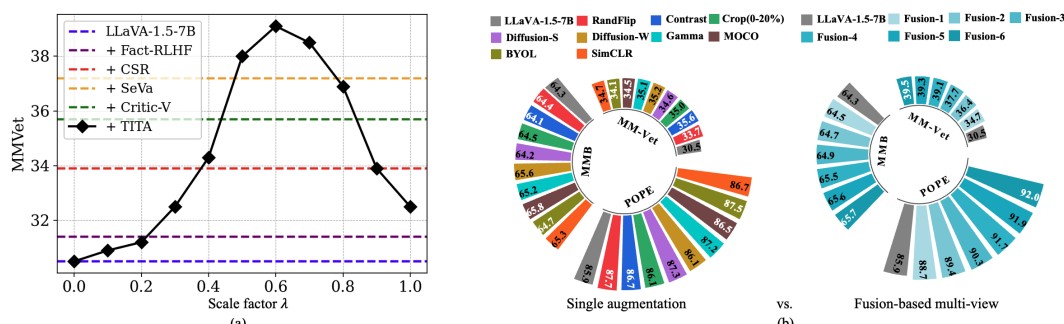

Figure 3: **Ablation studies on reward integration and reward modeling:** (a) MMVet accuracy under different scale factor $\lambda$ in Equation 8. TITA achieves optimal performance at $\lambda = 0.6$. (b) Comparison of single-view versus fusion-based reward modeling. Fusion-based multi-view preference construction consistently improves performance across MMVet, MMB, and POPE benchmarks.

combinations, TITA provides reward-guided token-level alignment. TITA achieves consistently stronger results across hallucination and reasoning benchmarks, more detailed in Appendix B.2.

## 4.3 ABLATIONS

**Ablation of scale factor $\lambda$.** Figure 3(a) reports the effect of $\lambda$ on MMVet. The black diamonds denote our method, and the dashed curves correspond to baseline results. As $\lambda$ increases from 0 to 0.6, MMVet improves from 30.3% to 39.0%, yielding a gain of 8.7 percentage points. At the optimal value $\lambda$=0.6, TITA surpasses the strongest baseline (SeVa) by about 1.6%, and exceeds Critic-V, CSR, Fact-RLHF, and the LLaVA-1.5-7B base model by 3.2%, 5.1%, 7.6%, and 8.7%, respectively. When $\lambda$ continues to increase, performance declines, likely because excessive reliance on the reward model reduces generation diversity or fluency. These results indicate that moderate values of $\lambda$ (approximately 0.5 to 0.7) strike a favorable balance between preference alignment and generation quality. Consistency of the peak region across tasks, as shown in Appendix C, further indicates that $\lambda$ is not highly sensitive and transfers reliably across evaluation settings.

**Ablation of reward model.** We assess the reward modeling strategy through two settings: (1) constructing preference pairs using a single image augmentation, and (2) constructing winners via our fusion-based approach. The left panel of Figure 3(b) shows that using a single augmentation (e.g., *RandFlip*, *Contrast*) leads to modest gains over the baseline—for example, *Contrast* and *Diffusion-W* improve MMVet by 3.1% and 2.7%. Although these augmentations provide useful preference signals, their limited semantic variation yields inconsistent improvements on MMB and POPE. In contrast, the right panel of Figure 3(b) shows that our fusion-based construction, which aggregates multiple augmented responses into a stronger winner, yields consistent improvements across benchmarks. As the number of fused responses increases (from *Fusion-1* to *Fusion-6*), performance steadily rises, reaching gains up to 8.6% on MMVet and 6.7% on POPE. These findings demonstrate the importance of stronger contrastive pairs and validate the effectiveness of multi-view fusion for reward modeling.

**Quantitative validation of $y_w$.** To further verify the superiority of fusion-based winners ($y_w$) over original responses ($y_l$), we using GPT-4o-2024-08-06 as the evaluator. Evaluation sets are constructed from TextVQA and OCRVQA, where each $(I, q)$ is paired with $y_w$ and $y_l$. As shown in Table 4, $y_w$ achieves significantly higher win rates (97.3% on TextVQA, 85.1% on OCRVQA), while $y_l$ is rarely preferred. These results provides strong quantitative evidence for adopting $y_w$ as the preferred winner in reward modeling.

Table 4: Quantitative comparison between fusion-based winners ($y_w$) and original responses ($y_l$).

| **Dataset** | $y_w$ win rate | $y_l$ win rate | Tie rate |
|---|---|---|---|
| TextVQA | 97.30% | 0.44% | 2.26% |
| OCRVQA | 85.12% | 2.95% | 11.93% |

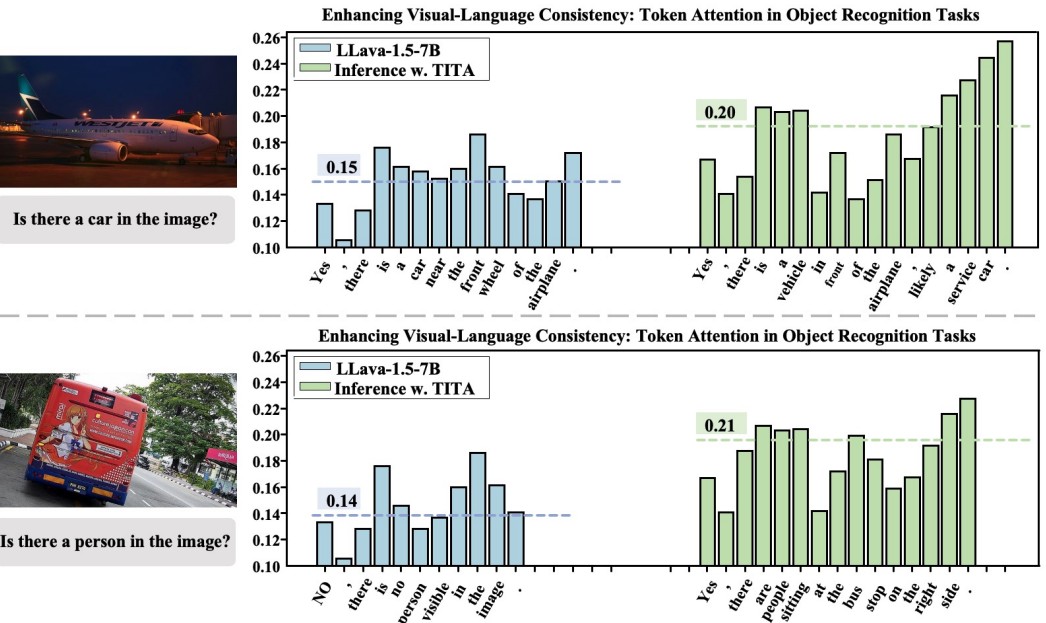

Figure 4: Visualization of response-token attention over visual features on the POPE benchmark. Compared to the baseline LLaVA-1.5-7B, `TITA`-guided inference produces higher and more focused attention weights on visually grounded tokens.

### 4.4 GENERATION EXAMPLES

To qualitatively assess the alignment improvements of `TITA`, we provide comparative generation examples in hallucination-prone scenarios. Figure 4 compares outputs from the baseline LLaVA-1.5-7B and the same model with `TITA` -guided inference on the POPE benchmark. The baseline often generates descriptions that reference objects or attributes absent from the input image, whereas the `TITA`-guided output remains consistent with the visual evidence, illustrating improved grounding.

To further understand this effect, we visualize response-token attention over visual features. The baseline shows diffuse or irrelevant attention, frequently neglecting salient regions of the image. In contrast, `TITA` yields sharper and semantically aligned attention distributions, suggesting stronger integration of visual cues into the decoding process. These qualitative observations complement the quantitative results in Section 4.2, demonstrating that `TITA` can reduce hallucinations and strengthen visual grounding at a fine-grained level without requiring original model retraining.

## 5 CONCLUSION

We introduced `TITA` , a lightweight inference-time framework for token-level alignment in VLMs. Unlike training-time alignment, it does not require finetuning or modifying the base model, and unlike supervised approaches, it avoids reliance on human-labeled token-level data. Instead, `TITA` transforms sparse sequence-level rewards into dense autoregressive signals, enabling fine-grained hallucination suppression directly during decoding. This is achieved by deriving implicit preference signals from log-probability ratios between a reward model and the target model, with a token-mapping mechanism ensuring compatibility across heterogeneous tokenizers. Experiments on three representative LVLM families (LLaVA, Qwen2.5-VL, DeepSeek-VL2) and twelve benchmarks demonstrate that `TITA` consistently reduces hallucinations, improves multimodal reasoning accuracy, and maintains low computational cost. Taken together, these results establish token-level inference-time alignment as an efficient and scalable paradigm for building reliable VLMs.

**Limitations.** While `TITA` relies on a reward model, we explicitly mitigate bias through log-probability ratio calibration and self-supervised preference construction, and the consistent improvements across 12 benchmarks indicate that residual bias has minimal practical impact.

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

# A  THEORETICAL JUSTIFICATION FOR LOG-PROBABILITY REWARD IN VLMS

In this subsection, we provide a theoretical justification for using the log-probability form $\log \pi_r(y \mid q, I)$ as a general parameterization of reward functions in preference-based learning for VLM. Here, the input $x = (q, I)$ encodes a query prompt $q$ and a corresponding image $I$. Modeling reward in this multimodal context poses unique challenges due to the entangled semantics of linguistic and visual inputs. We demonstrate that, under the Plackett-Luce model and its special case, the Bradley-Terry model, the log-likelihood $\log \pi_r(y \mid q, I)$ retains the full representational capacity of the reward function class—up to an equivalence relation that preserves both preference structures and the resulting optimal policy.

**Theorem I.**  Let $\mathcal{R}$ denote the class of reward functions consistent with the Plackett-Luce model over multimodal input $(q, I)$. Then, for every $r \in \mathcal{R}$, there exists a probability distribution $\pi_r(y \mid q, I)$ such that the log-probability reward $\log \pi_r(y \mid q, I)$ belongs to the same preference equivalence class as $r$. Moreover, this parameterization is unique within each equivalence class.

This result implies that using the autoregressive likelihood $\log \pi_r(y \mid q, I)$ as a surrogate reward function in VLMs is not merely an approximation but a complete and expressive formulation under the Plackett-Luce framework. Despite the complexity of multimodal grounding—where visual evidence and linguistic instructions jointly influence the response—the log-probability form preserves the full range of expressible preferences encoded by reward functions in $\mathcal{R}$.

To formalize this claim, we first define equivalence classes of reward functions based on the preference distributions they induce.

**Lemma.**  (Adapted from (Rafailov et al., 2024)) Under the Plackett-Luce or Bradley-Terry model, two reward functions $r_1(q, I, y)$ and $r_2(q, I, y)$ are equivalent if they induce the same pairwise preference probabilities over responses:

$$P(y \succ y' \mid q, I) = \frac{\exp(r(q, I, y))}{\exp(r(q, I, y)) + \exp(r(q, I, y'))}$$

Furthermore, any pair of equivalent reward functions leads to the same optimal policy in constrained reinforcement learning settings.

*Proof.*  Let $r(q, I, y) \in \mathcal{R}$ be an arbitrary reward function. Define its normalized variant via the softmax transformation:

$$\hat{r}(q, I, y) := \log \frac{\exp(r(q, I, y))}{\sum_z \exp(r(q, I, z))} = r(q, I, y) - \log \sum_z \exp(r(q, I, z))$$

The corresponding conditional distribution is:

$$\pi_r(y \mid q, I) = \frac{\exp(r(q, I, y))}{\sum_z \exp(r(q, I, z))},$$

and hence $\log \pi_r(y \mid q, I) = \hat{r}(q, I, y)$.

We now show that $\hat{r}(q, I, y)$ and $r(q, I, y)$ belong to the same preference equivalence class. Observe that the transformation introduces only a constant shift:

$$r(q, I, y) - \hat{r}(q, I, y) = \log \sum_z \exp(r(q, I, z)),$$

which is independent of $y$. Therefore, the pairwise preference between any two outputs remains unchanged:

$$\frac{\exp(r(q, I, y))}{\exp(r(q, I, y)) + \exp(r(q, I, y'))} = \frac{\exp(\hat{r}(q, I, y))}{\exp(\hat{r}(q, I, y)) + \exp(\hat{r}(q, I, y'))}.$$

Since the preference structure is preserved, the same ranking over outputs is induced, and thus the same optimal policy is obtained when optimizing under such preferences. This confirms that $\log \pi_r(y \mid q, I)$ is a faithful representative of the equivalence class defined by $r(q, I, y)$. $\qquad\square$

**Theorem II.** All reward equivalence classes can be represented with the parameterization $\log \pi_r(y|q, I)$ for some probablity distribution $\pi_r(y|q, I)$.

*Proof Sketch.* Take any reward function $r(q, I, y)$. Consider the following reward function

$$\hat{r}(q, I, y) := \log \frac{\exp r(q, I, y)}{\sum_z \exp r(q, I, z)}.$$

First, $\hat{r}(q, I, y)$ is consistent with the parameterization $\log \pi_r(y|q, I)$ with $\pi_r(y|q, I) = \frac{\exp r(q,I,y)}{\sum_z \exp r(q,I,z)}$. Second, since $r(q, I, y) - \hat{r}(q, I, y) = \log \sum_z \exp r(q, I, z)$ does not depend of $y$, $\hat{r}(q, I, y)$ and $r(q, I, y)$ are equivalent. Therefore, $\hat{r}(q, I, y)$ is a member of the equivalence class of $r(q, I, y)$ with the desired form, and we do not lose any generality in our reward model from the proposed parameterization. □

## B  EXPERIMENTAL DETAILS

### B.1  EVALUATION BENCHMARKS

LLaVA-Bench (In the wild) (Liu et al., 2024b): A challenging benchmark of 60 diverse tasks designed to evaluate models in naturalistic settings. It specifically tests visual instruction-following and question-answering capabilities in real-world scenarios, offering insights into practical applicability.

MM-Vet (Yu et al., 2023a): A comprehensive evaluation suite comprising 128 diverse tasks that assess six core visual-language capabilities. This benchmark uniquely combines mathematical reasoning, logical inference, and visual knowledge understanding, providing a rigorous test of multi-modal comprehension.

MM-Bench (Liu et al., 2025): A large-scale multi-modal benchmark with 4.7K samples, focusing on visual knowledge and reasoning capabilities. This dataset provides a balanced assessment of both factual knowledge and analytical reasoning in multi-modal contexts.

POPE (Li et al., 2023d): A specialized benchmark containing 8,440 samples designed to evaluate model hallucination. It specifically tests models' ability to provide accurate Yes/No responses about object presence in images, serving as a critical measure of visual grounding reliability.

MME (Yin et al., 2023): A benchmark with 14 tasks assessing perception and cognition in LVLMs, challenging interpretative and analytical skills.

SEED (Li et al., 2023b): A benchmark designed to evaluate the generative comprehension capabilities of large vision-language models (LVLMs). It includes an extensive dataset of 19K multiple-choice questions with precise human annotations, spanning 12 distinct evaluation dimensions that cover both spatial and temporal understanding across image and video modalities.

ScienceQA (Lu et al., 2022): A multimodal benchmark crafted to evaluate and diagnose the multi-hop reasoning abilities and interpretability of AI systems within the science domain. It features an extensive dataset of approximately 21k multiple-choice questions, spanning a broad spectrum of scientific topics and supplemented with detailed answer annotations, associated lectures, and explanations.

GQA (Hudson & Manning, 2019): A dataset specifically engineered for advanced real-world visual reasoning, utilizing scene graph-based structures to generate 22 million diverse, semantically-programmed questions. It incorporates novel evaluation metrics focusing on consistency, grounding, and plausibility, thereby establishing a rigorous standard for vision-language task assessment.

VisWiz (Gurari et al., 2018): A visual question answering (VQA) dataset derived from naturalistic settings, featuring over 31,000 visual questions. It is distinguished by its goal-oriented approach, with images captured by blind individuals and accompanied by their spoken queries, along with crowdsourced answers.

MMStar (Chen et al., 2024b): A benchmark of 1,500 test samples designed to address issues of low vision–language alignment and potential training-data leakage. It is carefully curated and spans 6 core capability areas and 18 fine-grained evaluation axes.

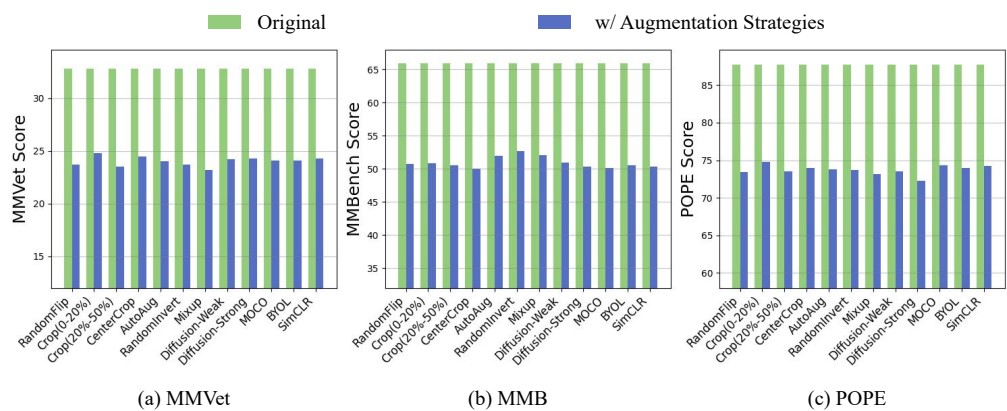

Figure 5: Comparison of 12 data augmentation strategies applied to LLaVA-1.5, including various geometric and color transformations as well as contrast learning enhancement methods. By analyzing these methods, the goal is to find the combination that best improves the performance of LVLMs.

CHAIR (Rohrbach et al., 2018): A well-established benchmark for evaluating object hallucination in image captioning tasks, with two variants: $\text{CHAIR}_i$ and $\text{CHAIR}_s$, which assess hallucination at the instance and sentence levels, respectively. we randomly sampled 500 images from the COCO (Lin et al., 2014) validation set and evaluated object hallucination using the CHAIR metric. Note that a lower CHAIR score indicates fewer hallucinations, which implies better alignment between the captions and the actual content of the images.

$$\text{CHAIR}_i = \frac{\text{Number of hallucinated objects}}{\text{Number of all mentioned objects}},$$

$$\text{CHAIR}_s = \frac{\text{Number of captions with hallucinated objects}}{\text{Number of all captions}}.$$

### B.2 EXPERIMENTAL SETUP

**Image augmentation strategies** We implement three effective image-side augmentation strategies to generate diverse responses from our model. By applying these techniques to the original images, we produce multiple distinct responses which are then synthesized into a comprehensive final output. This approach enhances model robustness by introducing controlled variations in visual input while maintaining semantic consistency. The augmentation strategies include:

- Crop($s_{\min}, s_{\max}$): Crop the image from minimum scale to the maximum scale ($s_{\min} = 0.2, s_{\max} = 0.5$ in our paper).
- Diffusion-S (Strong): Applies gaussian noise with 500 diffusion steps, creating significant but controlled perturbation.
- Diffusion-W (Weak): Introduces gaussian noise with 200 diffusion steps, offering a more moderate level of visual distortion.
- Contrast: Enhances image contrast by a factor of 2, accentuating visual boundaries and feature differences.
- Gamma: Performs gamma correction at a value of 0.8, lightening dark regions in the image. (Note that gamma values above 1 make shadows darker, while values below 1 make dark regions lighter).

**Impact with Augmentation Strategies** To assess the impact of augmentation strategies, we analyzed 12 widely used techniques (Chen et al., 2020; Grill et al., 2020; He et al., 2020) (Figure 5). We found that overly aggressive methods (e.g., strong diffusion noise) hindered feature learning, while overly simple ones (e.g., random flipping) offered limited gains. Accordingly, we adopted a balanced combination of three effective augmentations with the original images.

Table 5: Performance breakdown on MMVet, MMBench, and POPE benchmarks, covering subskills, multilingual understanding, and hallucination robustness.

| Model | MMVet | | | | | | | MMBench | | POPE | | | |
|---|---|---|---|---|---|---|---|---|---|---|---|---|---|
| | All | rec | ocr | know | gen | spat | math | en | cn | All | rand | pop | adv |
| LLaVA-1.5-7B | 30.5 | 35.7 | 21.9 | 17.7 | 19.7 | 24.7 | 7.7 | 64.3 | 58.3 | 85.9 | 89.5 | 86.7 | 81.7 |
| + Fact-RLHF | 31.4 | 36.5 | 22.7 | 18.1 | 20.9 | 32.3 | 7.7 | 63.4 | 56.8 | 81.5 | 86.5 | 83.9 | 83.0 |
| + CSR | 33.9 | 37.2 | 23.3 | 21.9 | 24.5 | 27.7 | 7.7 | 65.5 | 59.4 | 86.8 | 89.4 | 87.4 | 83.6 |
| + SeVa | 37.2 | 40.2 | 29.9 | 21.8 | 23.9 | 34.3 | 7.7 | 65.6 | 59.2 | 86.7 | 89.4 | 87.1 | 83.6 |
| + Critic-V | 35.7 | 37.6 | 28.1 | 21.0 | 22.5 | 28.5 | 7.7 | 64.0 | 58.5 | 86.5 | 88.1 | 86.4 | 83.5 |
| + TITA (Ours) | 39.1 | 44.8 | 31.2 | 30.7 | 34.5 | 36.0 | 7.7 | 65.5 | 59.2 | 91.7 | 92.6 | 93.0 | 90.2 |
| LLaVA-1.5-13B | 35.4 | 38.9 | 32.2 | 23.3 | 24.8 | 29.7 | 24.8 | 67.7 | 63.6 | 85.9 | 89.6 | 86.5 | 82.0 |
| + Fact-RLHF | 32.6 | 41.2 | 28.9 | 22.8 | 23.7 | 34.1 | 25.2 | 64.7 | 58.0 | 86.7 | 89.4 | 87.5 | 82.5 |
| + CSR | 37.8 | 41.0 | 32.5 | 24.6 | 30.1 | 32.8 | 24.8 | 68.8 | 64.5 | 87.3 | 89.4 | 88.1 | 82.2 |
| + SeVa | 41.0 | 45.4 | 32.8 | 32.4 | 36.7 | 37.0 | 25.4 | 68.7 | 64.8 | 87.4 | 90.5 | 89.0 | 82.7 |
| + Critic-V | 39.2 | 39.5 | 30.0 | 25.7 | 29.2 | 34.7 | 24.6 | 66.7 | 62.0 | 80.1 | 90.3 | 88.2 | 82.6 |
| + TITA (Ours) | 42.3 | 44.8 | 36.2 | 33.1 | 38.5 | 39.0 | 24.8 | 68.2 | 64.2 | 92.6 | 93.2 | 93.7 | 91.0 |

Table 6: Comparison of TITA with inference-time decoding methods.

| Model | Inference logits | CHAIR$_s$ ↓ | CHAIR$_i$ ↓ | POPE | MMVet |
|---|---|---|---|---|---|
| *Base Model: LLaVA-1.5-7B* | | | | | |
| Base | $\log \pi_\theta(y\|q, I)$ | 48.8 | 14.9 | 85.9 | 30.5 |
| + VCD (Leng et al., 2024) | $(1+\lambda)\log \pi_\theta(y\|q, I) - \lambda \log \pi_\theta(y\|q, \hat{I})$ | 28.1 | 11.0 | 86.3 | 32.9 |
| + M3ID (Favero et al., 2024) | $(1-\lambda)\log \pi_\theta(y\|q, I) + \lambda \log \pi_\theta(y\|q)$ | 27.1 | 6.4 | 88.0 | 36.2 |
| + MARINE (Zhao et al., 2024) | $(1-\lambda)\log \pi_\theta(y\|q, c, I) + \lambda \log \pi_\theta(y\|q, I)$ | 17.8 | 7.2 | 90.5 | 38.5 |
| + TITA (Ours) | $(1-\lambda)\log \pi_{\text{reward}}(y\|q, I) + \lambda \log \pi_\theta(y\|q, I)$ | 20.3 | 5.6 | 91.7 | 39.1 |

**Additiona Detail Results** Table 5 provides a detailed breakdown of performance across three representative benchmarks: MMVet, MMBench, and POPE. MMVet evaluates model capabilities across seven fine-grained categories, including reasoning (rec), OCR, knowledge, generation (gen), spatial understanding (spat), and math. MMBench is split into English (en) and Chinese (cn) subsets to assess multilingual general knowledge understanding. POPE focuses on hallucination detection, with evaluations under different conditions: random (rand), popular (pop), and adversarial (adv) prompts. These results highlight the consistent improvements brought by our method across diverse evaluation dimensions.

**Comparison with Rencen Decoding Method** We further examine the relationship between TITA and recent inference-time decoding optimization methods, including VCD (Leng et al., 2024), M3ID (Favero et al., 2024), and MARINE (Zhao et al., 2024) in the Table 6. These approaches adjust the decoding process by combining different conditional probability terms. While effective in certain cases, such heuristics lack explicit preference signals and therefore provide limited control over hallucination behavior.

### B.3 WHY VISUAL ATTENTION IN MIDDLE LAYERS IMPLIES HALLUCINATION

To understand why hallucinations emerge in VLMs and why TITA's decoding guidance is effective, we analyze how LLaVA-1.5-7B processes visual information during object-token generation. Prior work suggests (Li et al., 2023a; Zhu et al., 2023; Hurst et al., 2024; Shen et al., 2025) that VLMs rely heavily on linguistic priors, often before visual evidence is fully incorporated. We therefore examine (a) the visual attention ratios across layers and heads, and (b) the logit contribution of attention sublayers to real-object prediction. These diagnostics help identify where visual grounding happens, when language priors take over, and what goes wrong when hallucination occurs.

The further analysis in Fig. 6 reveals a clear two-stage processing pattern in LLaVA-1.5-7B. In the middle layers (5–18), the model consistently assigns higher attention to image tokens, indicating that these layers serve as a visual evidence accumulation stage. However, their direct contribution to the final output remains limited. In contrast, the upper layers (19–26) exhibit a sharp rise in logit contribution, reflecting a semantic refinement stage where the model converts accumulated representations into object-token predictions.

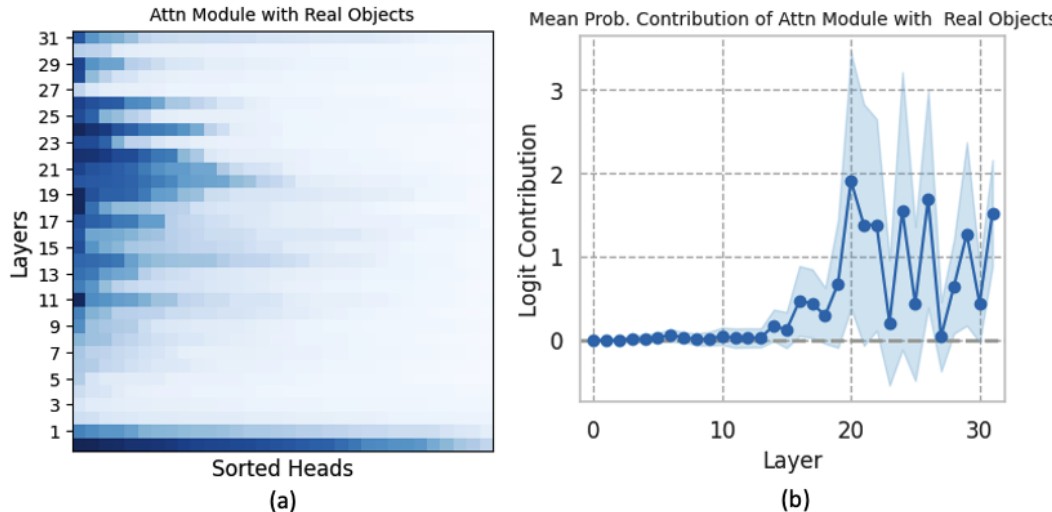

Figure 6: Visual Attention Dynamics Across Layers and Their Role in Grounded Object Generation. (a) Layer–head distribution of visual attention ratios for real object tokens in LLaVA-1.5-7B. Each row (layer) is sorted by attention ratio. (b) Mean logit contribution of attention sublayers to correct object-token prediction. Middle layers steadily gather visual information, while upper layers convert these representations into semantic predictions.

This layered structure explains why hallucinations occur: if the model fails to gather sufficiently strong visual grounding during the accumulation stage, the semantic refinement stage defaults to linguistic priors, leading to visually inconsistent outputs. TITA mitigates this failure mode by reinforcing visual-token attention precisely during the accumulation stage, ensuring that the refinement stage builds on reliable visual information rather than textual bias. These results provide quantitative evidence for the mechanism by which TITA reduces hallucinations.

## C  CASE STUDY: PLUG-AND-PLAY INTEGRATION

Our approach follows a plug-and-play paradigm, where a lightweight task-specific reward model guides a large-scale pre-trained language model during inference, without requiring fine-tuning or architectural modification of the target model. This modularity allows easy adaptation across domains and tasks. As illustrated in Figure 7, the reward model is first trained on domain-specific data, then used at inference time to inject task-aware preferences by influencing the token selection process through reward-weighted logits. This setup preserves the original capabilities of the base model while introducing fine-grained control from the auxiliary reward model.

A potential challenge in this plug-and-play setup is the mismatch between the tokenizers of the reward model and the target model. To ensure compatibility, we adopt a logits mapping strategy during inference. Specifically, at each decoding step $[t]$, we first obtain the top-$k$ tokens from the reward model's output distribution $\pi_r(y_{[t]} \mid x, y_{<[t]})$. These token IDs are decoded into text using the reward model's tokenizer. The resulting strings are then re-encoded using the target model's tokenizer to identify the corresponding token(s) in the target vocabulary. The reward scores from the original top-$k$ tokens are mapped to the re-encoded tokens, and the resulting distribution is aligned with the target model's vocabulary. Finally, the mapped reward logits are interpolated with the target model's original logits to form a reward-aware distribution for sampling. This mechanism enables effective reward transfer across models with different tokenization schemes, preserving the modularity and generality of our approach.

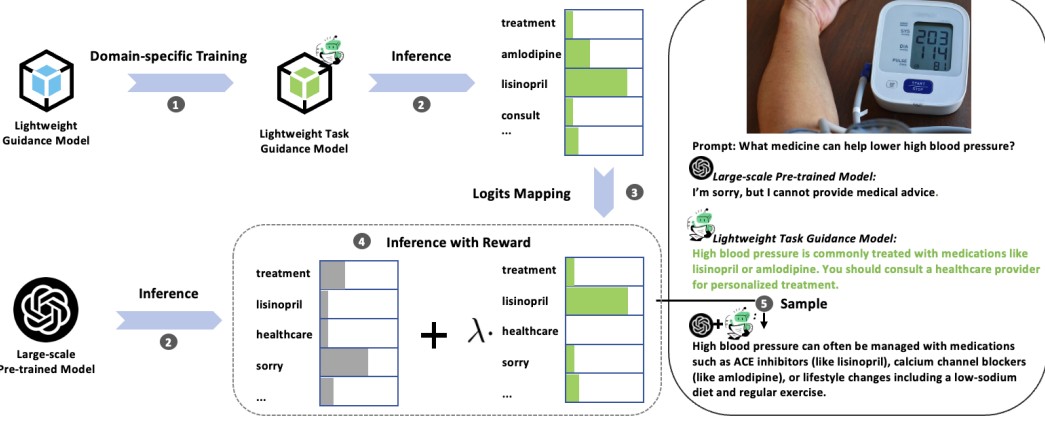

Figure 7: Token-level reward guidance using a lightweight model. Mapped reward logits are combined with the target model's logits to enable plug-and-play task adaptation without modifying the base model.

