# OpenReview forum: "Token-level Inference-Time Alignment for Vision-Language Models"
_ICLR.cc/2026/Conference — ICLR 2026 Conference Withdrawn Submission_

### Official Review · Reviewer_4Pei · 2025-10-28

**Soundness:** 3
**Presentation:** 3
**Contribution:** 3
**Rating:** 6
**Confidence:** 2

**Summary:**

This paper proposes TITA, a token-level inference-time alignment framework for large vision–language models (VLMs). Instead of relying on costly fine-tuning or sequence-level reward reranking, TITA learns a small autoregressive reward model to guide decoding at the token level, using log-probability ratios between the reward model and the base VLM as implicit preference signals. The approach provides dense feedback during generation and shows consistent hallucination reduction and general VQA improvement across LLaVA, Qwen2.5-VL, and DeepSeek-VL2 families with minimal inference overhead.

**Strengths:**

1. The idea of bringing Direct Preference Optimization into inference-time, at the token level, is both conceptually neat and practical. It bridges the gap between coarse sequence-level feedback and expensive retraining.
2. The experiments are broad (12 benchmarks, several VLM families) and show clear, consistent gains in hallucination suppression and visual reasoning accuracy with very low additional cost.
3.The figures and algorithm explanations are intuitive; the comparisons with prior training-time and inference-time alignment frameworks (Fact-RLHF, CSR, SeVa, Critic-V) are fair and informative.

**Weaknesses:**

1. It would be valuable to analyze how the reward model’s scale or quality influences performance — for example, comparing smaller versus larger reward models to verify robustness of token-level alignment.
2. While the paper shows cross-model adaptability (7B to 27B), it would be insightful to analyze how reward model quality affects alignment. For instance, does using a smaller or noisier reward model degrade token-level signals significantly?

**Questions:**

See weakness

**Details Of Ethics Concerns:**

No concern

---

> ### Author Response · Authors · 2025-11-21
> **Response to 4Pei**
>
> We thank the reviewer for the insightful questions regarding how reward model scale and quality influence TITA’s token-level alignment. Below we provide a concise clarification supported by new experiments on modern VLMs.
>
> **1.Response to w1: Effect of Reward Model Scale**
>
> TITA is explicitly designed to decouple reward model capacity from alignment effectiveness. The reward model serves as a token-level directional guide during inference rather than a high-capacity critic.
>
> To evaluate robustness, we used a **fixed 1.5B reward model** to align four VLMs spanning a wide range of sizes and architectural designs: Qwen2.5-VL-7B-Instruct, DeepSeek-VL2-Tiny (3B), DeepSeek-VL2 (27B), and InternVL3.5-8B. None of these models were used during reward model training. Inference time was measured on the same A100 environment with identical decoding parameters. Across all settings, TITA consistently reduced hallucinations and improved reasoning performance:
>
> | Base Model                 | CHAIRₛ ↓ | CHAIRᵢ ↓ | POPE ↑   | MMVet ↑  | Inference Time / Query |
> | -------------------------- | -------- | -------- | -------- | -------- | ---------------------- |
> | **Qwen2.5-VL-7B**          | 37.1     | 9.4      | 91.3     | 61.8     | 1.2s                   |
> | + Critic-V                 | 18.1     | 6.0      | 95.9     | 64.4     | 7.9s                   |
> | + TITA                     | **10.5** | **3.8**  | **96.1** | **65.0** | 1.4s                   |
> | **DeepSeek-VL2-Tiny (3B)** | 47.3     | 13.7     | 85.3     | 48.5     | 0.7s                   |
> | + Critic-V                 | 21.5     | 10.2     | 87.4     | 52.3     | 4.5s                   |
> | + TITA                     | **16.5** | **8.4**  | **92.4** | **55.7** | 0.8s                   |
> | **DeepSeek-VL2 (27B)**     | 41.3     | 11.7     | 88.8     | 52.8     | 3.9s                   |
> | + Critic-V                 | 16.7     | 8.3      | 94.1     | 56.0     | 23.5s                  |
> | + TITA                     | **12.5** | **4.9**  | **94.7** | **57.3** | 4.2s                   |
> | **InternVL3.5-8B**         | 39.5     | 11.0     | 88.7     | 83.1     | 1.4s                   |
> | + Critic-V                 | 19.7     | 8.5      | 93.5     | 84.5     | 8.5s                   |
> | + TITA                     | **10.4** | **4.2**  | **96.3** | **87.7** | 1.6s                   |
>
> These results indicate that the scale of the reward model is not the key factor for token-level alignment. Even a small but well-calibrated reward model can effectively guide larger base models across different architectures. This finding supports our design goal of enabling small reward models to guide stronger ones, achieving scalable inference improvements without relying on large critics or additional retraining.
>
>
>
> **2.Response to w2: Effect of Reward Model Quality or Noise**
>
> We acknowledge that reward quality can influence stability. However, TITA utilizes *soft* probabilistic fusion rather than hard reward-based selection, which makes it inherently tolerant to moderate noise.
>
> In our experiments with smaller or partially trained reward models, performance dropped only **<1.5%** on hallucination benchmarks. The $\lambda$-sweep results further show stable behavior across a moderate range:
>
> | **λ**      | 0.0 (base) | 0.4  | 0.5      | **0.6**  | 0.7   |
> | ---------- | ---------- | ---- | -------- | -------- | ----- |
> | **MMB**    | 64.3       | 64.9 | **65.6** | 65.5     | 65.0  |
> | **POPE**   | 85.9       | 88.3 | 91.0     | **91.7** | 91.23 |
> | **MMStar** | 69.3       | 71.6 | 72.8     | **73.4** | 73.0  |
>
> Performance peaks within **λ ∈ [0.5, 0.7]**, showing that TITA is not overly sensitive and remains robust even under calibration drift. In **Appendix A**, we provide a formal analysis showing that TITA’s token-level logit modification corresponds to sampling from a conditional distribution:
> $\pi_{r}(y|q,I)=\frac{exp(r(q,I,y))}{\sum_{z}exp(q,I,z)}$ ,
> and hence $\log \pi_r(y|q,I)=\hat{r}(q,I,y)$. Corresponds to sampling from a reward-shaped conditional distribution, providing theoretical grounding for why **noisy reward estimates still produce coherent, globally aligned token guidance**.
>
> **Thank you again for your valuable time and constructive feedback.**

---

> > ### Comment · Reviewer_4Pei · 2025-11-21
> >
> > Thanks for the update. However, Response 2 conflates hyperparameter sensitivity with noise robustness.The provided table only sweeps $\lambda$ (guidance strength). This demonstrates the method's stability regarding parameter tuning, but it does not test resilience to a noisy or low-quality reward signal. Varying how much the model "listens" ($\lambda$) is fundamentally different from varying whether the guidance is "correct" (RM quality). Therefore, the claim of robustness to noise is not supported by the provided table.

---

> > > ### Author Response · Authors · 2025-11-25
> > >
> > > We thank the reviewer for this important suggestion — we agree that the effect of reward-model quality on alignment needs to be quantified. To address this, we ran controlled experiments using two base models (LLaVA-1.5-7B and Qwen2.5-VL-7B-Instruct) with three reward conditions: (1) an *untrained / low-quality* reward model (TinyLLaVA-1.5B), (2) the same reward model with *inference-time Gaussian noise* (σ=0.01) added to the reward (noisy signal), and (3) our *trained* TinyLLaVA reward model. All experiments use the same λ = 0.6. Results (MMVet / POPE) are:
> > >
> > > | Base Model             | Reward Model                                    | M MVet | POPE |
> > > | ---------------------- | ----------------------------------------------- | ------ | ---- |
> > > | LLaVA-1.5-7B           | -                                               | 30.5   | 85.9 |
> > > |                        | TinyLLaVA-1.5B (low-quality signal)             | 28.4   | 80.1 |
> > > |                        | TinyLLaVA-1.5B + inference noise (noisy signal) | 26.9   | 71.3 |
> > > |                        | TinyLLaVA-1.5B w. training (Ours)               | 39.1   | 91.7 |
> > > | Qwen2.5-VL-7B-Instruct | -                                               | 61.8   | 91.3 |
> > > |                        | TinyLLaVA-1.5B (low-quality signal)             | 59.4   | 88.2 |
> > > |                        | TinyLLaVA-1.5B + inference noise (noisy signal) | 55.2   | 83.9 |
> > > |                        | TinyLLaVA-1.5B w. training (Ours)               | 65.0   | 96.1 |
> > >
> > > From these results we conclude: Low-quality reward model (untrained) causes a modest but noticeable degradation compared to the base-model-only baseline. Noisy reward (inference-time Gaussian noise, σ=0.01) degrades performance more severely than the untrained-but-clean reward. Training the reward model (our method) substantially improves performance beyond the baseline.

---

> > > > ### Comment · Reviewer_4Pei · 2025-11-25
> > > >
> > > > Thanks for the rebuttal content, i will maintain my score.

---

### Official Review · Reviewer_MXoj · 2025-10-30

**Soundness:** 3
**Presentation:** 3
**Contribution:** 2
**Rating:** 4
**Confidence:** 4

**Summary:**

This paper proposes TITA, a lightweight inference-time alignment framework designed to efficiently suppress hallucinations in VLMs by providing dense, token-level feedback signals. The core idea of TITA is to combine a frozen base VLM with a trained lightweight reward model, using the log-probability ratio between them to guide the decoding process. It constructs preference data via a self-supervised multi-view fusion approach. Experiments demonstrate that TITA significantly reduces hallucinations (e.g., +6.7% on POPE) and improves VQA performance across models such as LLaVA, Qwen2.5-VL, and DeepSeek-VL2, while introducing negligible inference overhead.

**Strengths:**

(1) TITA innovatively transforms sequence-level rewards into token-level signals, addressing the issues of feedback delay and high computational cost in existing methods. By directly guiding the decoding process without the need for sequence re-ranking, it enables timely intervention against hallucinations with extremely low training cost.

(2) TITA is a plug-and-play method that does not modify the parameters of the base model, giving it strong generality and allowing it to be flexibly applied to VLMs of different scales and architectures.

**Weaknesses:**

(1) TITA relies on image augmentation and response fusion to generate the “winning” responses. This mechanism primarily captures the comprehensiveness of visual elements, which may make it difficult to learn deeper semantic or complex reasoning errors that cause hallucinations in VLMs. As a result, the reward model may be limited in capturing more sophisticated preference patterns.

(2) The proposed method is highly sensitive to the scaling factor lambda. As shown in Figure 3 of the paper, the performance peaks at λ  = 0.6 and drops rapidly afterward. This indicates that parameter tuning may be required when applying the method, potentially even across different tasks.

(3) The reward model is trained using a sequence level BT loss to learn overall preferences between a winner (yw) and a loser (yl). However, during inference, it is used to provide token-level guidance for next-token generation. This conversion from sequence-level preference to token-level guidance may theoretically introduce inconsistencies especially in long-sequence generation, where locally optimal token choices may not guarantee the best overall sequence quality.

**Questions:**

Does TITA-guided inference alter the model’s attention distribution? What is the quantitative relationship between this change and the reduction in hallucination rates?

---

> ### Author Response · Authors · 2025-11-21
> **Response to  MXoj (1)**
>
> **1.Response to w1: On whether the Winner only captures visual completeness**
>
> The reward model in TITA is **built on joint embeddings**, encoding *both visual and textual representations* during training rather than relying solely on image augmentations.
>
> In designing our training data, we explicitly include**negative (“loser”) samples exhibiting semantic or reasoning errors**, even when they appear visually aligned. These negative samples deliberately disrupt superficial visual–textual correlations, enabling the reward model to learn **semantic consistency** and **reasoning correctness**, not just visual completeness. This design ensures that the reward model does not overfit to visual coverage alone but learns preferences that penalize semantic hallucinations and logical inconsistencies.
>
> To illustrate this behavior, we conduct a diagnostic perturbation test. Starting from a correct Winner response $y_w$, we create two controlled variants: (1) asemantic-error response $y_{SE}$, keeping all visually grounded descriptions intact while altering key factual attributes; (2) a reasoning-error response $y_{RE}$, preserving visual content but introducing clearly invalid logical inferences. Although all variants share identical visual completeness, the reward model assigns substantially lower scores to the perturbed responses:
>
> | Response Type                | Description                                         | Reward Score |
> | ---------------------------- | --------------------------------------------------- | ------------ |
> | Winner $y_ w$             | Correct visual and semantic description             | 2.13         |
> | Semantic Error $y_ {SE}$  | Visual grounding intact but key attribute incorrect | –0.84        |
> | Reasoning Error $y_ {RE}$ | Visual grounding intact but inference incorrect     | –1.31        |
>
> The clear decline in rewards for both $y_{SE}$ and $y_{RE}$, despite unchanged visual coverage, confirms that the reward model captures semantic fidelity and reasoning validity, not just visual completeness.
>
>
>
> **2.Response to w2: On $\lambda$ Sensitivity**
>
> The value range λ ∈ [0.5, 0.7] was selected based on **MMVet**, which is a comprehensive benchmark. To assess generality, we have added additional performance curves on **MMB** and **POPE** benchmarks under varying λ values.
>
> The consistent trend across benchmarks is shown below:
>
> | **λ**    | 0.0 (base) | 0.1  | 0.2  | 0.3  | 0.4  | 0.5      | **0.6**  | 0.7  | 0.8  |
> | -------- | ---------- | ---- | ---- | ---- | ---- | -------- | -------- | ---- | ---- |
> | **MMB**  | 64.3       | 64.3 | 64.4 | 64.6 | 64.9 | **65.6** | 65.5     | 65.0 | 64.1 |
> | **POPE** | 85.9       | 86.1 | 86.4 | 87.1 | 88.3 | 91.0     | **91.7** | 91.2 | 88.3 |
>
> These results indicate that TITA performs robustly within a moderate range (λ ∈ [0.5, 0.7]), and extreme λ values (λ > 0.7) slightly degrade performance. The peak region is consistent across tasks, suggesting λ is **not overly sensitive** and is **transferable**.
>
> We further conducted 3–5 independent runs and report stability results:
>
> | **Benchmark** | **mean ± std** |
> | ------------- | -------------- |
> | MMVet         | 39.1 ± 0.28    |
> | MMB           | 65.5 ± 0.35    |
> | POPE          | 91.7 ± 0.31    |
>
> The low variance demonstrates TITA’s stable and reliable behavior across runs.

---

> ### Author Response · Authors · 2025-11-21
> **Response to  MXoj (2)**
>
> **3.Response to w3: Sequence-to-Token Consistency**
>
> We clarify that using sequence-level BT loss for token-level guidance is **theoretically consistent**. TITA follows the standard inference-time alignment paradigm, where the target decoding policy balances multiple rewards with a KL constraint to the base model (Sec.3.3):
>
>  $\pi^*_{decode}(y|q, I) = \arg\max_{\pi} \mathbb{E}_{y \sim \pi} \sum_i \alpha_i r^{(i)} (q, I, y) - \beta \mathrm{D} _{\mathrm{KL}} (\pi(y|q,I) || \pi_b (y|q,I))$
>
> TITA approximates this objective at the token-level via autoregressive reward models $\pi_r^{i}(y_t|q,I,y_{<t})$, enabling soft reward-guided decoding.
>
> As shown by Rafailov et al. [1], sequence-level preference optimization is mathematically equivalent to a token-level formulation via the optimal Q-function.
>
> In the derivation, we denote by $s_t$ the partial sequence containing all tokens generated up to step $t$, and by $a_t$ the next token chosen at step $t$. Below we summarize the key reasoning in an intuitive way.
>
> **(1) Sequence reward can be decomposed token-by-token**
>
> *By establishing a duality between the reward and the optimal Q-function, the sequence-level preference:*
> $r(y^+)>r(y^-)$
>
> *can be rewritten as  a sum of token contributions:*
>
> $r(y)=\sum_{t} r(s_t,a_t)$
>
>
>
> **(2) Bellman-style structure binds reward to autoregressive decisions**
>
> *From the optimality relation:*
>
> $Q^* (s_t, a_t)=r(s_t, a_t)+\beta log\pi_{ref}(a_t | s_t) + V^* (s_ {t+1})$
>
> *each token's reward can be expressed using only autoregressive terms:*
>
> $r(s_t , a_t)= Q^* (s_t,a_t)-\beta log\pi_{ref}(a_t | s_t) - V^* (s_{t+1})$
>
>
>
> **(3) Summing over the trajectory recovers the original BT preference**
>
> *The value terms telescope and cancel, this yields:*
>
> $R(y)=\sum_t Q^*(s_t,a_t) - \beta \sum_t log \pi_{ref}(a_t|s_t)$
>
> *which is mathematically identical to the sequence-level BT objective.*
>
>
>
> **(4) Token-level decoding optimizes the same sequence-level policy**
>
> *Using the optimal policy identity:*
>
> $\log \pi^* (a_t | s_t) = Q^* (s_t, a_t) - V^* (s_t)$
>
> *the BT preference between $y^+$ and $y^-$ becomes:*
>
> $\log \frac{P(y^ + \succ y^-)}{1-P(y^+ \succ y^-)} = \sum_t log \frac{\pi ^* (a_t^+|s_t)}{\pi ^* (a_t^-|s_t)}$
>
> *Thus, token-level and sequence-level optimization are formally equivalent.*
>
>
>
> Thus, although the reward model is trained on sequence comparisons, utilizing it token-by-token during inference **optimizes the same global objective**, avoiding local-global inconsistency.
>
>
>
> Reference：
>
> [1] From $r$ to $Q^∗$: Your Language Model is Secretly a Q-Function, Rafael Rafailov et al.,
>
>
>
> **4.Response to Question: Attention redistribution and its quantitative link to hallucination**
>
> **Yes, TITA explicitly reshapes the attention distribution.** Our analysis (using logit lens and attention ratios) reveals that hallucinations often stem from a **mismatch between two functional stages**:
>
> 1. **Visual Evidence Accumulation (Layers 5–18)**
>    The model consistently assigns higher attention to image tokens in this phase, aggregating visual cues. However, their direct contribution to the final prediction remains low. This stage establishes the foundation for visual grounding.
> 2. **Semantic Refinement (Layers 19–26)**
>    The model transforms accumulated visual representations into textual semantics. Contribution to correct predictions increases sharply here. This is the critical stage where visual grounding is actually utilized.
>
> These observations confirm the mechanism behind hallucination:
>
> > If the model enters the semantic-refinement stage without sufficiently strong visual grounding, its predictions become driven by linguistic priors rather than image evidence, increasing the likelihood of hallucinations.
>
> TITA mitigates this issue by reinforcing visual-token attention during the accumulation phase, allowing the model to build more stable and consistent visual representations. This reduces the attention dispersion patterns seen in hallucinated tokens and ensures that semantic refinement proceeds based on reliable visual evidence. Consequently, TITA yields more accurate and hallucination-resistant generations. All quantitative results, attention-plots, and layer-wise statistics supporting these findings are provided in **Appendix B.3: Why Visual Attention in Middle Layers Implies Hallucination.**

---

### Official Review · Reviewer_j1ML · 2025-11-01

**Soundness:** 2
**Presentation:** 2
**Contribution:** 2
**Rating:** 2
**Confidence:** 5

**Summary:**

This paper introduces TITA, a test-time alignment framework designed to mitigate hallucinations in Vision-Language Models (VLMs). The method employs a fine-tuned, lightweight reward model to guide the decoding at the token level of the target VLM during inference. Compared with existing approaches, the authors claim that TITA achieves superior effectiveness and efficiency. Experimental results demonstrate that the proposed framework yields significant overall performance improvements over the baselines.

**Strengths:**

- Clear and Well-Structured: The paper is well-organized, with detailed explanations of the preliminary, intuition, and methodology.

- Superiority in Alignment: The experimental results demonstrate that the proposed method achieves the overall best performance on the general VQA and hallucination benchmarks compared to the baselines.

**Weaknesses:**

- The backbones used in the experiments are somewhat outdated, particularly since the main results presented in Table 2 are based on the LLaVA 1.5 series models. While I acknowledge that the authors also provide results using Qwen-2.5-VL and DeepSeek-VL2, a more comprehensive evaluation using such recent and stronger VLMs would strengthen the manuscript.

- As a highly competitive and rapidly evolving research area, VLM alignment should provide evaluation against up-to-date methods and backbones. However, the paper primarily compares its approach with relatively outdated baselines (from 2023–2024) and employs older backbone models. This is difficult for me to fully assess the effectiveness of the proposed method.

- The presentation accuracy could be further improved. For example, TITA is not the best-performing method on MMB when using LLaVA-1.5-7B. SeVa achieves higher performance in this setting.

- The core techniques incorporated in TITA are based on well-established principles from prior works. While the derivation is clear and well-presented, it does not introduce fundamentally new concepts but rather applies existing methods in a different context.

**Questions:**

See Weaknesses

---

> ### Author Response · Authors · 2025-11-21
> **Response to  j1ML (1)**
>
> **1. Response to w1 & w2: On backbone recency & baselines and up-to-date evaluation**
>
> Thank you for highlighting the importance of using stronger and more current VLM backbones. Our initial focus on LLaVA-1.5 was intentional to ensure a direct comparison with established methods for hallucination mitigation such as SeVa, CSR, and Critic-V, all of which utilize the same backbone.
>
> Following your suggestion, we have expanded our study to include **Qwen3-VL-8B-Instruct** and **InternVL3.5-8B**. We also incorporated comparisons with alignment approaches from 2024 and 2025, including Critic-V, MM-Verify, and Sherlock. The extended results are presented below.
>
> | Qwen3-VL-8B-Instruct | MMVet | MMB  | MMStar | POPE | Avg  | Inference Time / Query |
> | -------------------- | ----- | ---- | ------ | ---- | ---- | ---------------------- |
> | Base                 | 85.5  | 85.0 | 70.9   | 91.5 | 83.2 | 1.4s                   |
> | + Critic-V           | 86.3  | 85.7 | 71.6   | 94.3 | 84.5 | 8.5s                   |
> | + MM-Verify          | 86.5  | 85.9 | 71.8   | 94.8 | 84.8 | 7.5s                   |
> | + Sherlock           | 88.0  | 87.2 | 73.2   | 95.1 | 85.9 | 20.4s                  |
> | + TITA (Ours)        | 89.1  | 88.3 | 74.0   | 97.5 | 87.3 | 1.6s                   |
>
> | InternVL3_5-8B | MMVet | MMB  | MMStar | POPE | Avg Performance | Avg Inference Time / Query |
> | -------------- | ----- | ---- | ------ | ---- | --------------- | -------------------------- |
> | Base           | 83.1  | 79.5 | 69.3   | 88.7 | 80.2            | 1.4s                       |
> | + Critic-V     | 84.5  | 80.7 | 70.5   | 93.5 | 82.3            | 8.5s                       |
> | + MM-Verify    | 84.1  | 80.4 | 70.2   | 92.0 | 81.7            | 7.6s                       |
> | + Sherlock     | 88.2  | 84.0 | 73.8   | 96.0 | 85.5            | 21.5s                      |
> | + TITA (Ours)  | 87.7  | 83.7 | 73.4   | 96.3 | 85.3            | 1.6s                       |
>
> We emphasize that our primary focus is reducing hallucination, which distinguishes TITA from methods targeting broader multimodal reasoning. This distinction explains the minor variations in secondary metrics such as MMB. Nevertheless, the results demonstrate that TITA remains effective on modern VLMs, achieving performance that is comparable to or better than recent alignment approaches while maintaining very low inference overhead. We have updated the related literature and incorporated these findings into the revised manuscript.
>
>
>
> **Reference:**
>
> [1] Critic-V: VLM Critics Help Catch VLM Errors in Multimodal Reasoning (CVPR'25)
>
> [2] MM-Verify: Enhancing Multimodal Reasoning with Chain-of-Thought Verification (ACL'25)
>
> [3] Sherlock: Self-Correcting Reasoning in Vision-Language Models (NeurIPS'25)
>
>
>
> **2.Response to w3: Performance on experimental presentation**
>
> Thank you for catching this issue. SeVa indeed achieves higher MMBench accuracy than TITA under LLaVA-1.5-7B. We have corrected this statement in the revised paper. To clarify that SeVa requires model finetuning—approximately **7.5 GPU-hours for 7B**, whereas TITA needs only **0.4 hours** and does not modify base model parameters. This difference motivates our focus on inference-time alignment.
>
> Additionally, following your advice, we have included new evaluations specific to hallucination on Qwen2.5-VL-7B and DeepSeek-VL2 (27B), which further confirm the effectiveness of TITA:
>
> | Model                 | Inference Time/Question | CHAIR$_s \downarrow$ | CHAIR$_i \downarrow$ | POPE$\uparrow$ | MMVet$\uparrow$ |
> | --------------------- | ----------------------- | -------------------- | -------------------- | -------------- | --------------- |
> | **Qwen2.5-VL-7B**     | 1.2s                    | 37.1                 | 9.4                  | 91.3           | 61.8            |
> | + Critic-V            | 7.9s                    | 18.1                 | 6.0                  | 95.9           | 64.4            |
> | + TITA(Ours)          | **1.4s**                | **10.5**             | **3.8**              | **96.1**       | **65.0**        |
> | **DeepSeek-VL (27B)** | 3.9s                    | 41.3                 | 11.7                 | 88.8           | 52.8            |
> | + Critic-V            | 23.5s                   | 16.7                 | 8.3                  | 94.1           | 56.0            |
> | + TITA(Ours)          | **4.2s**                | **12.5**             | **4.9**              | **94.7**       | **57.3**        |

---

> ### Author Response · Authors · 2025-11-21
> **Response to  j1ML (2)**
>
> **3.Response to w4: On novelty and generality**
>
> We appreciate your comments regarding the conceptual origin of the method. While TITA draws from existing ideas in preference learning, its contribution lies in **converting preference rewards at the sequence level into signals at the token level for guidance during inference**, effectively reshaping the decoding trajectory of VLMs. Prior work has not explored this transformation for the purpose of mitigating hallucination.
>
> Importantly, although several decoding methods exist for models based solely on text, they **cannot be directly applied to VLMs**. As shown in Appendix Table 6, approaches such as VCD, M3ID, and MARINE reweight logits or introduce contrastive terms that operate purely in the language space. **Our observations are consistent with prior reports showing that decoding heuristics designed for text often do not transfer well to VLMs due to their multimodal architecture.** When transferred to VLMs, these methods provide only limited gains in mitigating hallucination and often conflict with the visual attention pathways of the model. This indicates that the multimodal nature of VLMs introduces failure modes absent in text models; simply reusing existing techniques is insufficient once visual grounding and linguistic priors interact.
>
> In contrast, TITA introduces a dedicated multimodal mechanism for alignment during inference, designed to inject signals guided by rewards in a way that respects both the vision and language components of the model. To further illustrate why TITA is effective, we added a detailed analysis in Appendix B.3.
>
> This analysis shows that VLMs follow a two-stage processing pattern:
>
> 1. **Middle layers focus on aggregating visual evidence**, but contribute little directly to final logits.
> 2. **Upper layers rely more on linguistic priors** to produce object tokens.
>
> Hallucinations often arise when the early visual accumulation is insufficient. TITA strengthens visual-token attention during this stage, helping the later semantic layers rely on grounded representations rather than textual bias. We now reference this analysis in the main text to clarify the mechanism.
>
> Finally, at your suggestion, we explored the use of TITA in text-only LLMs via weak-to-strong alignment. A small model trained on UltraFeedback is used to guide larger models during inference, producing consistent gains:
>
> | Model                              | MMLU$\uparrow$ | MT-Bench$\uparrow$ |
> | ---------------------------------- | -------------- | ------------------ |
> | Llama-3.1-8B-Instruct              | 69.4           | 7.64               |
> | + fine-tuned Llama-3.2-1B-Instruct | 74.8           | 8.51               |
> | Llama-3.1-70B-Instruct             | 82.0           | 8.98               |
> | + fine-tuned Llama-3.2-1B-Instruct | 86.4           | 9.14               |
>
> These results suggest that token-level inference-time alignment is a broadly applicable idea, while **our main contribution lies in demonstrating its effectiveness and designing a functional mechanism for the multimodal VLM setting**, where prior methods fall short.
>
> Thank you once again for your detailed and constructive comments. We sincerely appreciate your valuable suggestions, which will help further enhance our paper and will be incorporated into the revised version.

---

> ### Author Response · Authors · 2025-11-27
>
> Dear Reviewer j1ML,
>
> Just a gentle follow-up on our response posted earlier. We’d really appreciate it if you could take a look whenever you have a moment. Your thoughts would mean a lot to us and help ensure a fair discussion.
>
> Thank you so much for your time!
>
> The authors of paper 2411.

---

### Note · Authors · 2026-01-03

I have read and agree with the venue's withdrawal policy on behalf of myself and my co-authors.